Journal of Machine Learning Research 1 (2021) 1-48          Submitted 4/00; Published 10/00

# Ensemble clustering for histopathological images segmentation using convolutional autoencoders

**Editor:** COMPAY 2021

## Abstract

Unsupervised deep learning using autoencoders has shown excellent results in image analysis and computer vision. However, only few studies have been presented in the field of digital pathology, where proper labelling of the objects of interest is a particularly costly and difficult task. Thus, having a first fully unsupervised segmentation could greatly help in the analysis process of such images. In this paper, many architectures of convolutional autoencoders have been compared to study the influence of three main hyperparameters: (1) number of convolutional layers, (2) number of convolutions in each layer and (3) size of the latent space. Different clustering algorithms are also compared and we propose a new way to obtain more precise results by applying ensemble clustering techniques which consists in combining multiple clustering results.

**Keywords:** Autoencoders, Ensemble clustering, Digital pathology

## 1. Introduction

Pathology is essential for the diagnosis evaluation and understanding of many underlying biological and physiological mechanisms. It is usually a visual evaluation by pathologists of a tissue sample using a microscope to identify its structural properties. Currently, the visual evaluation of microscopic specimens is largely an unassisted process, and the pathologist's accuracy is established through extensive training, comparative analysis, peer quality control and personal experience. However, this field has undergone several technological revolutions in recent years with the advent of virtual microscopy (conversion of glass slides into high-resolution images called Whole Slide Images - WSI), often referred to as "digital pathology". Thus, major efforts have been made to design image analysis tools, for example to identify basic biological structures (stroma, immune cells, tumour, etc.), in order to make it easier for doctors to (semi-)automate the interpretation of slides.

Meanwhile, automatic image analysis algorithms have recently made extraordinary progress, particularly with the advent of the deep learning methods introduced by Lecun *et al.* LeCun et al. (2015). Indeed, the performances of these methods have exploded in recent years, allowing the detection, classification and segmentation of objects of interest in images with very high precision. But most of these approaches operate in supervised mode, i.e. they require many examples in order to provide an effective model. However, obtaining quality annotations on histopathological images remains very costly. For example, in the field of colorectal cancer WSI segmentation, Qaiser *et al.* Qaiser et al. (2016) proposed a method based on persistent homology to classify tumour and non-tumour patches from

Hematoxylin & Eosin stained histology images. To train their system, more than 18000 annotated patches were needed.

At the same time, unsupervised approaches have shown their interest in many applications for image analysis, such as remote sensing Liang et al. (2018); Mei et al. (2019). Recently, they have also been applied to histopathological WSI analysis. In particular, in Yamamoto et al. (2019), the authors describe an unsupervised approach for extracting interesting information from WSI that obtains better accuracy than human for prognostic prediction of prostate cancer recurrence.

In this paper, we are interested in automatic segmentation in order to quickly extract regions of interest (tumours for example) to make a more precise analysis of these areas only. However, only few approaches on fully unsupervised segmentation of WSI have been proposed. The first attempt to segment regions of interest from WSI without any prior information or examples has been performed in Khan et al. (2013). The authors highlight tissue morphology in breast cancer histology images by calculating a set of Gabor filters to discriminate different regions. In Fouad et al. (2017), Fouad *et al.* use mathematical morphology to extract 'virtual-cells' (e.g. superpixels), for which morphological and colour features are calculated to then apply a consensus clustering algorithm to identify the different tissues in the image. More recently, a similar approach has been presented by Landini *et al.* Landini et al. (2019), adding a semi-supervised self-training classifier to the previous techniques that enhances the results at the cost of partial supervision.

All these approaches propose to cluster the image based on predefined features. However, deep learning approaches, particularly via autoencoding architectures, make it possible to avoid manual definition of features by calculating a condensed representation of the image in a latent space by applying convolutional filters. Unfortunately, as stated in Raza and Singh (2018), most applications of autoencoders in digital pathology were developed to perform cell segmentation or nuclei detection Xu et al. (2015); Hou et al. (2019), or stain normalisation Janowczyk et al. (2017). Therefore, we propose here to study the potential of these approaches for WSI tissue segmentation. The aim is to try to automatically identify clusters corresponding to each type of tissue in the WSI that could then be labelled by pathologists.

In this paper, we present a study on how convolutional autoencoders perform on WSI segmentation by comparing different approaches. First, different autoencoders architectures are compared to quantify the importance of hyperparameters of interest (number of convolutional layers, number of convolutions by layer and size of the latent space). Then, a multi-resolution approach using an ensemble clustering framework is evaluated, to see if such ensemble techniques could provide more accurate results.

## 2. Methods

### 2.1 Convolutional autoencoders for WSI clustering

In this section, we explore of the use of convolutional autoencoders to cluster WSI histopathological images. For this, we present several experiments to evaluate the importance of each hyperparameter.

As shown in Figure 1, a *Convolutional AutoEncoder* (CAE) is a deep convolutional neural network composed of two parts: an encoder and a decoder. The main purpose of

the CAE is to minimise a loss function $L$, evaluating the difference between the input and the output of the CAE (usually Mean Squared Error). Once this function is minimised, we can assume that the encoder part builds up a suitable summary of the input data, in the latent space, as the decoder part is capable of reconstructing an accurate copy of it from this encoded representation.

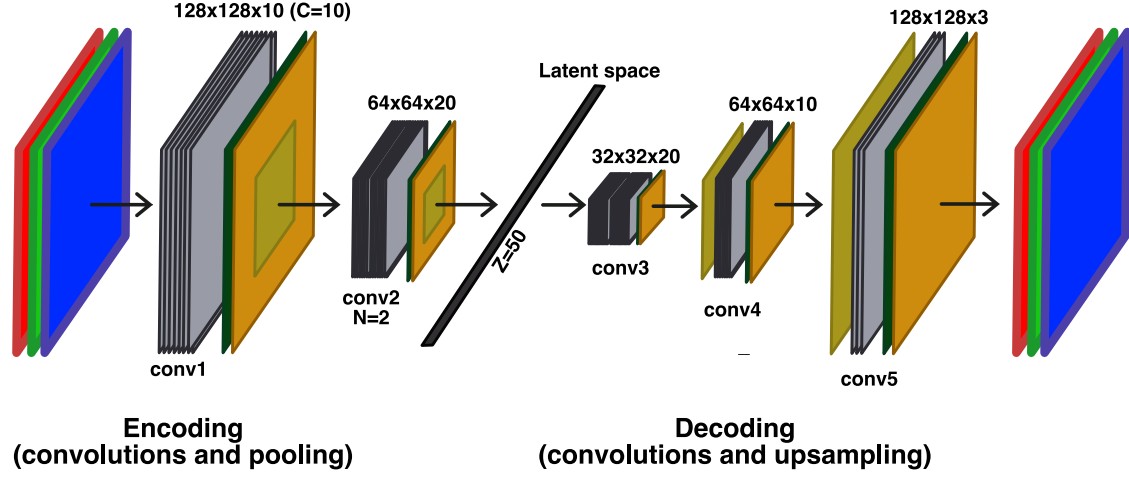

Figure 1: Architecture of a CAE with $N = 2$, $C = 10$ and $Z = 50$

The encoder is first constituted of the input layer (having the size of the input image) which is connected to $N$ convolutional layers of diminishing size, up to an information bottleneck of size $Z$, called the *latent space*. The bottleneck is connected to a series of $N$ convolutional layers of increasing size, until reaching the size of the input. This second part is called the decoder. Each convolution layer is composed of $C$ convolutions and is followed by three other layers: a batch normalisation, an activation function (ReLU) and a max pooling of size (2,2).

To perform the clustering, a trained CAE is used to encode each patch of the whole image. Then, this encoded representation of the patch (in the latent space) is given as the input of a clustering algorithm and a cluster is assigned to the patch.

We decided to evaluate the influence of the three hyperparameters $N$, $Z$ and $C$. For each one, different values were tested while fixing the two others ($N = 2$, $Z = 250$, $C = 10$). To evaluate the quality of the results, the *Adjusted Rand Index* ($ARI$) is calculated to compare the obtained clustering to the annotations of the expert. The *Rand Index* computes a similarity measure between two clusterings by considering all pairs of samples and counting pairs that are assigned in the same or different clusters in the predicted and true clusterings. The score is then normalised into the $ARI$ score by:

$$ARI = (RI - Expected\_RI)/(max(RI) - Expected\_RI) \qquad (1)$$

Values of the $ARI$ are close to 0 for random labelling independently of the number of clusters and samples, and exactly 1 when the clusterings are identical (up to a permutation).

Each CAE was trained over a set of 10,000 different patches randomly selected. As the result of both the clustering and the training of the CAE are non-deterministic, due

to a high sensitivity to the initial conditions, 10 autoencoders were trained and the results averaged for each hyperparameter value.

We also investigated the performance of several clustering algorithms, i.e Kmeans, Agglomerative clustering (AggCl), Gaussian mixture (GM) and also the not too deep clustering method (N2D) exposed in McConville et al. (2020). A clustering performed directly with the Kmeans algorithm on the raw data (without any data reduction by the CAE) has been calculated as a baseline to evaluate the benefit of encoding the data with the CAE.

## 2.2 Ensemble clustering

As exposed in Yamamoto et al. (2019), both micro-structures and macro-structures give different information. Pathologists also agree that identifying a single cell is way more difficult without its surrounding context and they always look at the WSI at lower magnification (to better capture the context) before zooming in at high magnification. Furthermore, in Alsubaie et al. (2018) an example of multi-resolution lung cancer adenocarcinoma classification using deep learning shows improvements in the overall accuracy.

Thus, we explored a way to improve the results by using an ensemble of clustering methods, each focusing on a different resolution. The objective is to merge low level information (context) with high level information (shape of the cells, etc.). For this, the consensus method proposed in Wemmert and Gançarski (2002) was used. This method is based on a the evaluation of the similarity between different clusterings and the definition of corresponding clusters. Then, a multi-view voting approach is computed to produce a single result representing all clusterings.

An example of the architecture of the approach is depicted in Figure 2.

We explored different configurations, but we only present the two most representative which highlight how the quality of the results can be improved by using ensemble clustering. The first configuration, $E_{\mathrm{multires}}$ is composed of three clustering algorithms (Kmeans) working on the latent space representation of the image obtained by different CAE trained at different resolutions: $10\times$ with 8 clusters, $5\times$ with 6 clusters and $5\times$ with 8 clusters. As the reconstructed image from the autoencoder seems to focus more on colour intensity than real structures, a second ensemble configuration has been tested. To add diversity and to force the final result to focus its attention more on the structure of the objects, a clustering working on a binary image (by thresholding the intensity of the initial image) has been computed. Thus, the second configuration ($E_{\mathrm{struct}}$) is composed of three clustering algorithms (Kmeans) with the following parameters: $5\times$ on the binary image with 6 clusters, $5\times$ on the binary image with 8 clusters and $10\times$ on the initial RGB image with 6 clusters.

## 3. Results

### 3.1 Data

Our study was performed on 8 WSI of Haematoxylin Eosin Saffron (HES) stained tissue extracted from a cohort of patients built within the scope of the AiCOLO project (INSERM/Plan Cancer) studying colon cancer. The images have been provided by Georges François Leclerc Centre (Dijon, France) and acquired from two different centres. An exam-

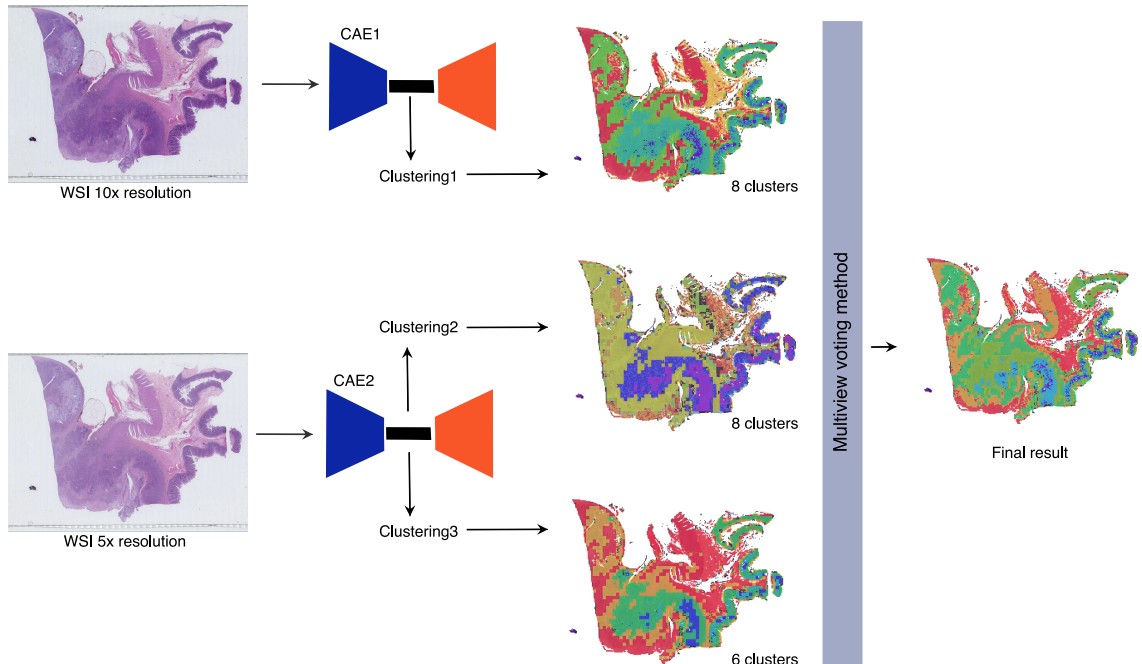

Figure 2: Architecture of the first ensemble configuration $E_{\mathrm{multires}}$: two CAE trained at different resolutions (10× and 5×) produce different latent representations that are clustered. The three resulting clusterings are then merged through the multi-view voting algorithm proposed in Wemmert and Gançarski (2002).

ple is given in Figure 3a. HES stain distinguishes cell nuclei in purple, from extracellular matrix and cytoplasm in pink.

All images have been acquired at 20× magnification (corresponding to 0.5 $\mu m$/pixel) but stored at several resolutions in a pyramidal format. The size of each image is around $90,000 \times 50,000$ pixels.

To train autoencoders, $10,000$ patches of size $128 \times 128$ pixels were randomly extracted at 10× resolution from all images (and 5× for the ensemble approach), as this seems to be the minimal amount of information required by human expert to classify the tissue. Meanwhile, sparse manual annotations of the five classes of tissue, tumour, stroma, outer layer mucosa (crypts of Lieberkuhn and connective tissue), immune cells, and necrosis, and two classes for background and artefacts (ink marks, etc.) have been performed by pathologists on the images (using CytomineMarée et al. (2016)), to be able to evaluate the relevance of the clustering.

## 3.2 Evaluation of all hyperparameters of the CAE

First, results obtained without using the latent space representation (see Table 1) are worse than all those obtained when clustering the encoded data. This confirms the interest of using a CAE for WSI clustering. As shown in Figure 4a, it appears that the number of convolutions in each layer of convolutions (hyperparameter $C$) does not greatly affects the

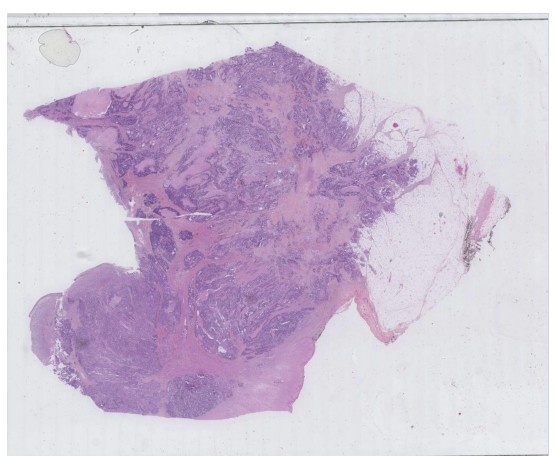 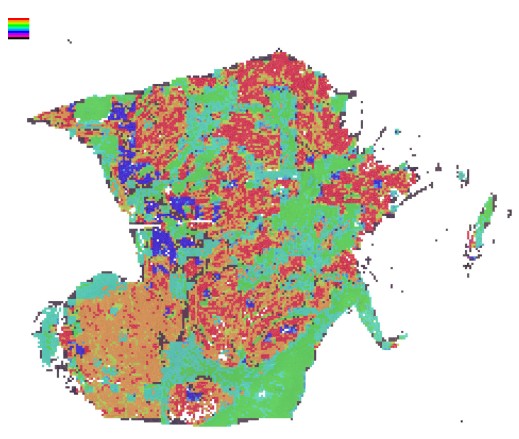

(a) Example of a WSI of colon tissue stained with HES (magnification: ×20, size: 97,920× 55,040 pixels)

(b) Example of clustering with 8 clusters (orange, red and blue clusters corresponding to tumour)

Figure 3: Example of a raw WSI (a) and a clustering result (b).

quality of the autoencoder as only a apart from a slight narrowing of the variability of the results. It's quite easy to figure out why: passed a certain number, additional convolution brings to few complementary information. Figure 4b shows the evaluation of the *ARI* with different number of convolution layers in the architecture. We can notice an increase of the quality index up to 4 layers and then a brutal drop at 5. This indicates clearly that too many convolutions (and poolings that downsample the information) reduce the information that can further not be properly processed.

Nonetheless, as seen in Figure 5, the latent space size $Z$, seems to greatly influence the pertinence of the CAE. Indeed, the *ARI* clearly grows as there is more space to encode the latent representation, as a more precise information can be stored. Also, the more information is present in the latent representation, the more classes can easily be differentiated. However, it is also clear that a too large latent space will not be able to summarise efficiently the information, and thus, will not help the clustering algorithm to discriminate the different tissues. Moreover, the larger the latent space, the more memory and time are needed to train the network.

### 3.3 Comparison of the CAE with the ensemble approach

As seen in the previous experiment, the *ARI* tends to give low scores because we only have very few annotations on each class of interest. So we decided to compute a second evaluation criterion based on the ability of the clustering to detect tumours areas in the image, as it is the main class of interest in our project. To associate the tumour class to a cluster, we calculated its tumour density (number of labelled tumour pixels / number of total labelled pixels in the cluster). All clusters having a density over 50% are kept as 'tumour', the others are labelled as 'not tumour'. Thus, two evaluation criteria have been

calculated on the results and are presented in Table 1 and Table 2: the *ARI* as in the previous experiment (see Eq.1) and the *Fscore* on the two-classes problem (tumour vs. not tumour) Van Rijsbergen (1979).

| | Raw data | Encoded data | | | | | |
|---|---|---|---|---|---|---|---|
| | **Kmeans** | **Kmeans** | **AggCl** | **GM** | **N2D** | $E_{\mathbf{multires}}$ | $E_{\mathbf{struct}}$ |
| Image 1 | 0.39 | **0.48** | 0.38 | 0.27 | 0.43 | 0.47 | 0.42 |
| Image 2 | 0.27 | 0.33 | 0.29 | 0.19 | 0.29 | 0.31 | **0.46** |
| Image 3 | 0.25 | 0.39 | 0.35 | 0.22 | 0.31 | 0.37 | **0.45** |
| Image 4 | 0.08 | 0.08 | 0.13 | 0.05 | **0.12** | 0.08 | 0.08 |
| Image 5 | 0.11 | 0.11 | 0.10 | 0.10 | 0.11 | 0.12 | **0.17** |
| Image 6 | 0.37 | 0.52 | 0.51 | 0.49 | 0.43 | 0.51 | **0.57** |
| Image 7 | 0.28 | 0.35 | 0.33 | 0.14 | 0.37 | **0.41** | 0.36 |
| Image 8 | 0.33 | 0.44 | 0.42 | 0.07 | 0.37 | 0.44 | **0.45** |

Table 1: Evaluation of the *ARI* of all clustering results obtained with the different methods.

| | Raw data | Encoded data | | | | | |
|---|---|---|---|---|---|---|---|
| | **Kmeans** | **Kmeans** | **AggCl** | **GM** | **N2D** | $E_{\mathbf{multires}}$ | $E_{\mathbf{struct}}$ |
| Image 1 | 0.89 | 0.96 | 0.94 | 0.73 | **0.97** | 0.96 | 0.88 |
| Image 2 | 0.68 | 0.63 | 0.62 | 0.43 | **0.68** | 0.66 | 0.62 |
| Image 3 | 0.76 | 0.87 | 0.85 | 0.76 | 0.88 | 0.87 | **0.91** |
| Image 4 | 0.48 | 0.50 | **0.61** | 0.51 | 0.55 | 0.54 | 0.48 |
| Image 5 | 0.65 | 0.64 | 0.60 | 0.62 | 0.65 | 0.65 | **0.72** |
| Image 6 | 0.68 | 0.75 | 0.72 | 0.67 | 0.76 | **0.77** | 0.75 |
| Image 7 | 0.68 | 0.73 | 0.75 | 0.61 | 0.74 | 0.76 | **0.84** |
| Image 8 | 0.63 | 0.71 | 0.70 | 0.44 | 0.69 | 0.75 | **0.75** |

Table 2: Evaluation of the *Fscore* of all clustering results obtained with the different methods.

## 4. Discussion

Classical methods applied on the latent space representation of the CAE tend to show acceptable results. However, both ensemble clustering configurations seem to be more efficient in finding coherent clusters corresponding to the classes of interest defined by the pathologists.

Among all the exposed methods, $E_{\text{struct}}$ seems to give the best results. It tends to confirm the importance of the shape of the objects on histopathological images. Furthermore, it shows that even if convolutional autoencoders aim at automatically finding the best features to encode images, they can also take advantage of pre-computed features for some specific tasks.

## 5. Conclusion

In this paper, we compared different configurations of convolutional autoencoders in the field of unsupervised learning for WSI histopathological image segmentation. For this, different CAE architectures have been compared to try to find the best configuration and to study the influence of each hyperparameter. Then, we proposed a new approach that uses ensemble clustering technique to take advantage of multiresolution information and structural features in the image. This confirms the importance of having diversity in an ensemble learning framework and that working at different resolutions at the same time can really improve the quality of the results.

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

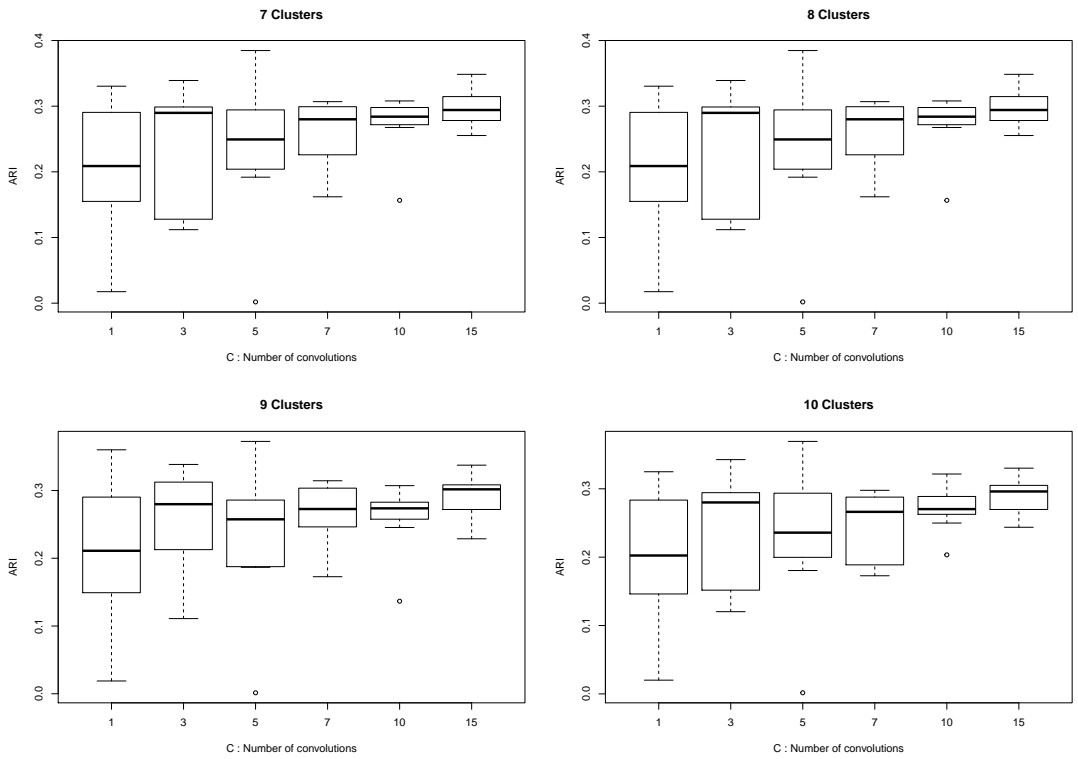

(a) Number of convolutions $C$ in each layer of convolutions ($N = 2$, $Z = 250$)

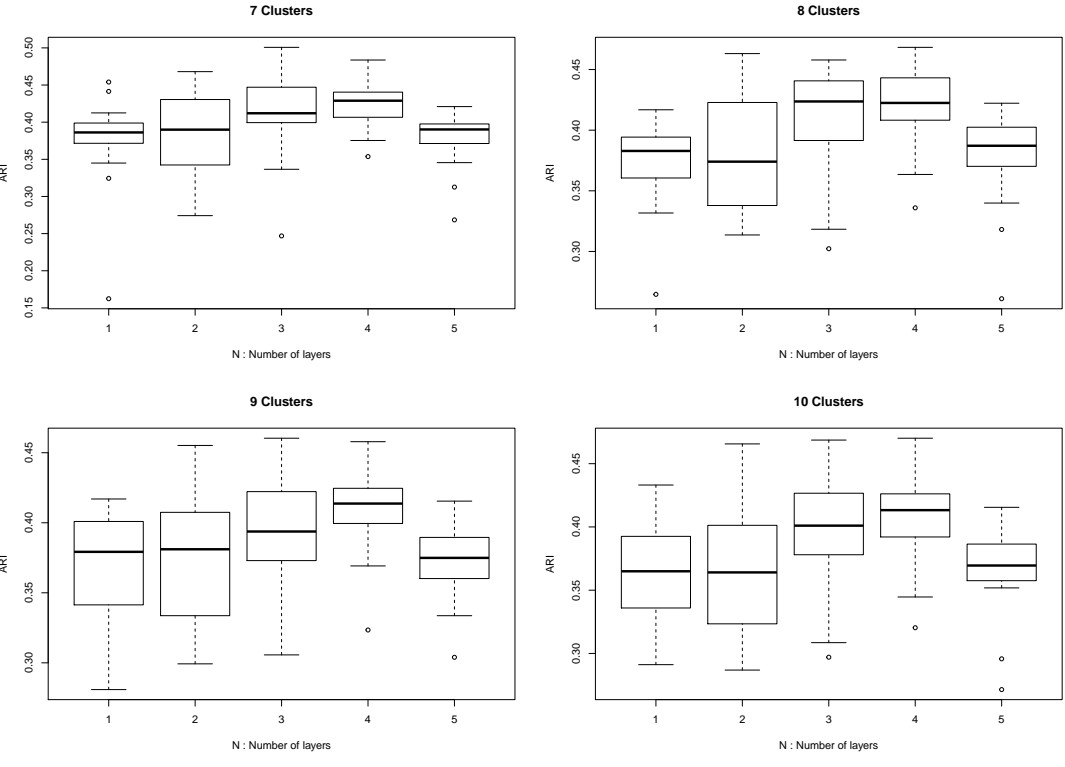

(b) Number of layers of convolutions $N$ ($Z = 250$, $C = 10$)

Figure 4: Evaluation of the ARI for the two main hyperparameters of the convolutions of the CAE comparing Kmeans clustering on 7, 8, 9 and 10 clusters.

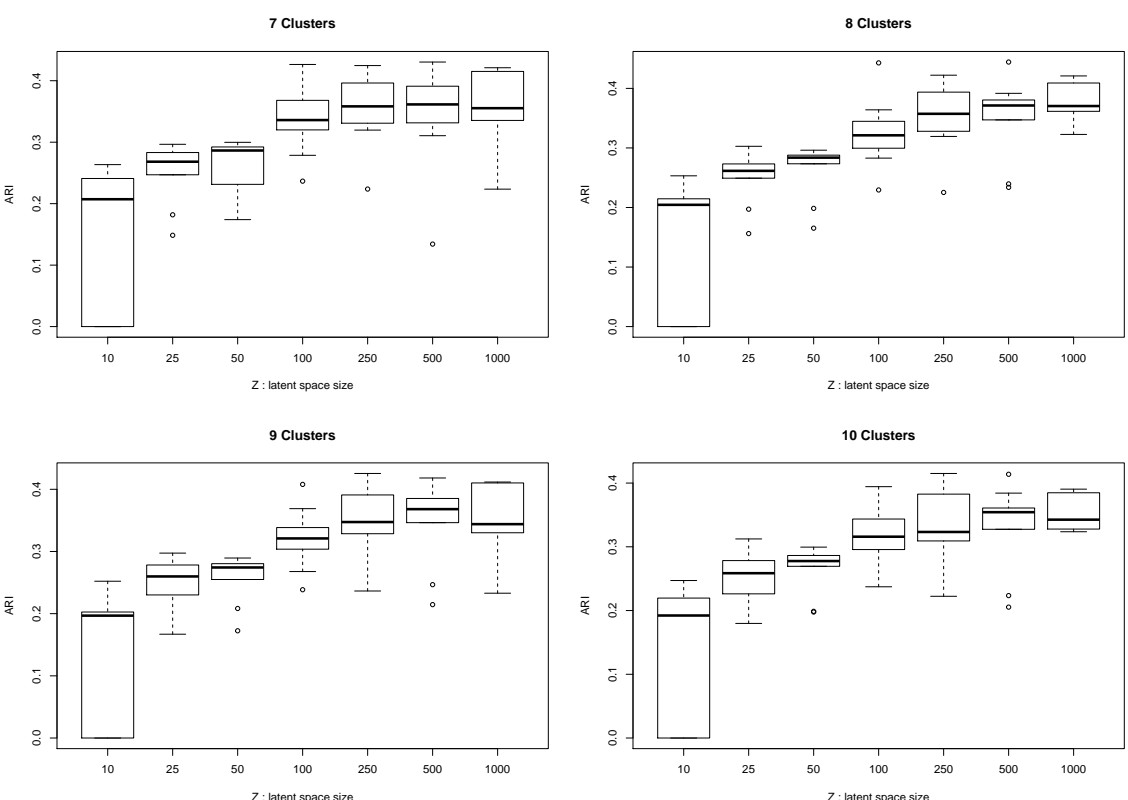

Figure 5: Evaluation of the *ARI* with different latent space sizes, comparing Kmeans clustering on 7, 8, 9 and 10 clusters.

