# OpenReview forum: "Ensemble clustering for histopathological images segmentation using convolutional autoencoders"
_MICCAI.org/2021/Workshop/COMPAY — Reject_

### Official Review · Reviewer_CmLq · 2021-08-05
**??**

**Rating:** 6
**Confidence:** 3

**Review:**

The authors present an unsupervised tissue segmentation approach for whole-slide images. They evaluate the influence of a number of parameters of the network and the use of ensemble clustering techniques. The methods are evaluated on a limited dataset with human annotations as gold standard, but the details of the manual gold standard are minimal.

The idea of the approach makes sense, in particular the notion of multiple scales for analysis. However, the results show only marginal improvement for the ensemble approach. This could very well be a result of the limited dataset and the relatively poor quality of the gold standard. It would be interesting to see outcomes for the approach on a more extensive set. I do find some of the methodological ideas worth investigating further.

The paper is clear and well structured. The writing does contain numerous small errors and typos, though. The style of the paper is rather chronological, in the sense that experiments or evaluation measures were added based on the first results. It is preferred to set up a study design and hypotheses prior to any experiments and to keep those fixed.

---

### Official Review · Reviewer_NAkw · 2021-08-22
**A step towards unsupervised tissue segmentation**

**Rating:** 4
**Confidence:** 4

**Review:**

Summary:

This paper investigated the effectiveness of combing convolutional autoencoders and clustering algorithms for tissue segmentation in histology images. The authors comparatively analysed the impact of different configurations of the autoencoder and different clustering methods on the classification performance.

Pros of the paper:
-	The paper paves the way for unsupervised tissue segmentation that can greatly save time and effort for pathologists.
-	The paper proposes comparisons among different parameter settings of the autoencoder and highlights the influence of latent space size.
-	The importance of aggregating features from multiple resolutions is appreciated.

Cons of the paper:
- Details of the loss function is missing. It has been suggested in [R1] that jointly optimizing MSE, Structural Similarity index and Mean Absolute Error can improve the performance of the autoencoder for tumour segmentation. They also proposed replacing the max-pooling operation with a convolutional layer with stride 2 to better reconstruct the image. I wonder if these configurations have been taken into account while designing the model architecture.
- The gap between the fully unsupervised method and the state-of-the-art supervised method is not shown. In the real application scenario, I suppose the method still requires some supervision to determine the type of tissues represented by the clusters. How does this approach compare with the classical classifiers in terms of the amount of annotation required?
- Figure 2 and 3 should include a map of manual annotations for evaluation.
- Features of patches grouped into distinct clusters need further investigation to clarify whether the autoencoder is able to capture characteristics specific to a tissue type.
- How is the number of clusters determined? And how is the multi-view voting algorithm applied in detail?
- For the F score evaluation on Image 2, Kmeans based on raw image data seems to outperform most of the clustering methods based on encoded features. Is this because the tumour region is well distinguishable by the colour intensity in the original image?

Writing quality:

The paper is well written except for a few typos, for example, “Pathology is essential for the diagnosis evaluation and understanding of many underlying biological and physiological mechanisms.” in page 1 (should be diagnostic); “does not greatly affects the quality of the autoencoder as only a apart from a slight narrowing of the variability of the results.” in page 6.

[R1] M. Roy et al., “Convolutional autoencoder based model HistoCAE for segmentation of viable tumor regions in liver whole-slide images,” Sci. Reports 2021 111, vol. 11, no. 1, pp. 1–10, Jan. 2021.

---

### Decision · Program_Chairs · 2021-08-25

Reject